# The Effect of pH and Buffer on Oligonucleotide Affinity for Iron Oxide Nanoparticles

Ekaterina Bobrikova [ID], Alexey Chubarov *[ID] and Elena Dmitrienko *[ID]

Institute of Chemical Biology and Fundamental Medicine, Siberian Branch of the RAS,
630090 Novosibirsk, Russia; e.bobrikova96@gmail.com

* Correspondence: chubarov@niboch.nsc.ru or chubarovalesha@mail.ru (A.C.); elenad@niboch.nsc.ru (E.D.);
Tel.: +7-913-763-1420 (A.C.); +7-913-904-1742 (E.D.)

**Abstract:** Magnetic $Fe_3O_4$ nanoparticles (MNPs) have great potential in the nucleic acid delivery approach for therapeutic applications. Herein, the formation of a stable complex of iron oxide nanoparticles with oligonucleotides was investigated. Several factors, such as pH, buffer components, and oligonucleotides sequences, were chosen for binding efficiency studies and oligonucleotide binding constant calculation. Standard characterization techniques, such as dynamic light scattering, zeta potential, and transmission electron microscopy, provide MNPs coating and stability. The toxicity experiments were performed using lung adenocarcinoma A549 cell line and high reactive oxygen species formation with methylene blue assay. $Fe_3O_4$ MNPs complexes with oligonucleotides show high stability and excellent biocompatibility.

**Keywords:** iron oxide nanoparticles; functionalization; coating; nanomedicine; DNA/gene delivery systems; toxicity; biostability





## 1. Introduction

Metal nanoparticles represent a significant doorway for various biomedical applications [1–7]. The use of nanoparticles as carriers for therapeutic molecules has been investigated to improve the therapeutic effect and avoid side effects. One of the approaches is to incorporate a magnetic core into the nanoparticle structure. Magnetic nanoparticles (MNPs) have become a vital nanomedicine material for drug delivery, biosensing, and biocatalysis [1–4,8–12]. MNPs can be directed around the body with a magnetic field, allowing for directed treatment, and drug/gene delivery strategy. Thus, MNPs provide therapy and diagnostic construction developments which are called theranostics.

Iron oxide MNPs are specifically considered a highly efficient contrast agent for magnetic resonance imaging (MRI) and hyperthermia therapy [1–3,11,13–19]. Iron oxide MNPs are particularly promising tags due to their high physical and chemical stability, cost-effectivity, and the optimal surface-to-volume ratio [20]. Among the iron oxide family, the most popular MNPs are $Fe_3O_4$, $\alpha$-, and $\gamma$-$Fe_2O_3$. Compared with other oxide nanoparticles, $Fe_3O_4$ has shown ferromagnetism characteristics generated by the spin magnetic moments of $Fe^{2+}$ and $Fe^{3+}$ ions [5]. However, $Fe_3O_4$ MNPs are not stable upon oxidation and possess high surface energy, leading to aggregation. Therefore, it is essential to consider the functionalization of such MNPs with polymers. The wrong functionalization leads to instability in biological liquids, toxicity, and high reactive oxygen species formation in cell lines and animal models [2,4,21–24]. The modified coatings improve the stability, aggregation properties, and biocompatibility of naked $Fe_3O_4$ MNPs. Indeed, the coating provides drug, ligand, DNA, or gene binding to the nanoparticle's surface. MNPs can be easily manipulated using an external magnetic field. which provides a magnetic field to nanoparticles for quick separation from the biological liquids and drugs/gene release at targets tissue.

Targeted nucleic acid delivery systems are particularly beneficial due to their unique properties to regulate gene expression, protein synthesis, and other fundamental processes in the cell [25–27]. Functionally integrating nucleic acids with nanoparticles display unique properties due to the synergistic activity of components. Magnetofection is an approach combining magnetic targeting an external field and nucleic acid delivery [28–32]. The magnetofection using MNPs encompass various nucleic acid types, such as DNA [33–35], small interfering RNA (siRNA) [35–40], micro RNA (miRNA) [41], and aptamers [42–45], have shown great potential in nanomedicine and fundamental investigations. However, the constructions conducted with nucleic acids represent a hard-obtained feature of multilayer nanocontainers. In our previous work [46], we have studied the possibility of oligonucleotide binding on gold nanoparticles. We examined oligonucleotides of different lengths, nucleotide composition, and the influence of other factors on non-covalent associates' formation.

Herein, we investigate the surface modification of iron oxide magnetic nanoparticles with various model sequence oligonucleotides to possess stabilized MNPs. The efficiency of [32]P-labeled oligonucleotides binding as a function of pH and binding constants was investigated. The stability of obtained MNPs was investigated in various buffers and pH. $\zeta$-potential measurements studied changes in particle surface. Transmission electron microscopy (TEM) and dynamic light scattering (DLS) were used for average particle size, size distribution, and morphology investigation. Toxicity experiments using lung adenocarcinoma A549 cell line and high reactive oxygen species formation experiments were performed. $Fe_3O_4$ MNPs complexes with oligonucleotides show high stability and excellent biocompatibility.

## 2. Results and Discussion

### 2.1. Synthesis and Characterization of MNP

Magnetic mixed iron oxide nanoparticles were synthesized by the coprecipitation method [47–49]. Particles are formed by mixing aqueous solutions of $Fe^{2+}$ and $Fe^{3+}$ iron salts in a ratio of 1:2, respectively, and successively changing the pH of the solution from acidic (adding HCl) to alkaline (adding $NH_3$), washing with perchloric acid at the final stage of synthesis. The synthesis scheme is shown in Figure 1.

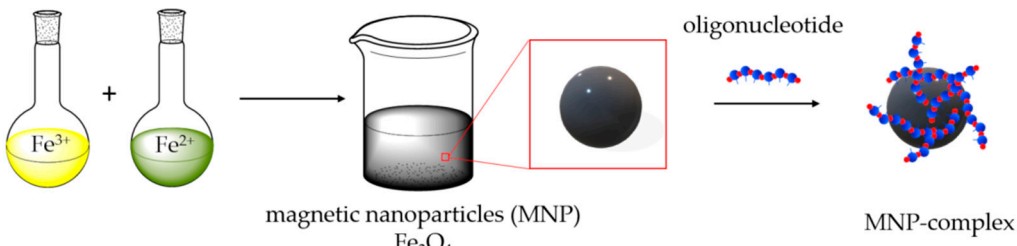

**Figure 1.** Representation of magnetic nanoparticles (MNPs) synthesis and its surface modification by oligonucleotides.

The resulting nanoparticles were characterized by TEM and DLS. The study of MNPs storage stability has shown that they are stable and do not form aggregates for six months (Supplementary Figures S1–S4). Figure 2 shows that the particles are monodisperse and do not exceed 20 nm in size. The DLS method determined the particle size to be 15.6 ± 2.2 nm (polydispersity index (PDI) = 0.296 ± 0.030, Figure 3). The actual diameter of nanoparticles is in a good correlation with DLS data and ranges from 12 to 17 nm (according to TEM images, Figure 2). The difference between DLS and TEM data is associated with the principle of measurement. DLS displays the characteristics of the hydrated particles and compounds in a solvent solution. $\zeta$-potential of MNP is 36.0 ± 0.9 mV, which indicates that MNPs have positively charged surfaces.

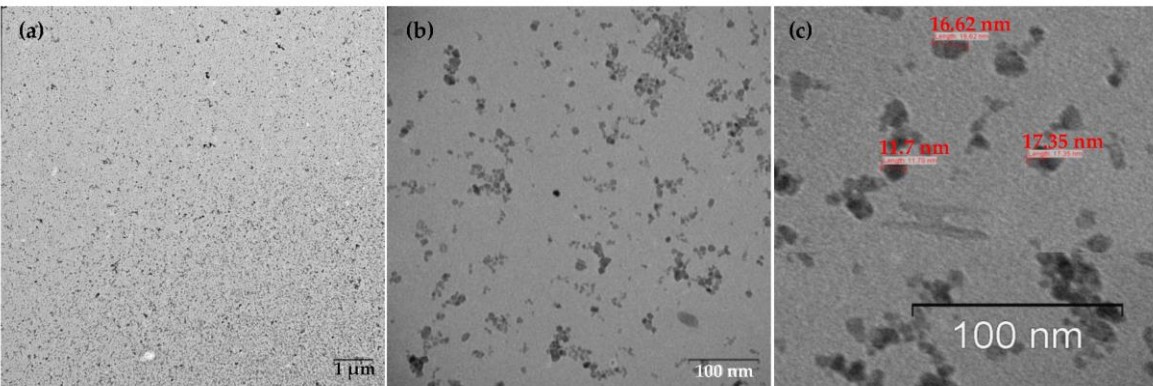

**Figure 2.** TEM images of MNPs. (**a**) Low magnification TEM image, the bar indicates one μm, (**b**) High magnification TEM image, the bar shows 100 nm. (**c**) TEM image reveals the MNPs size (~16–17 nm), monodisperse, and balls-like nature.

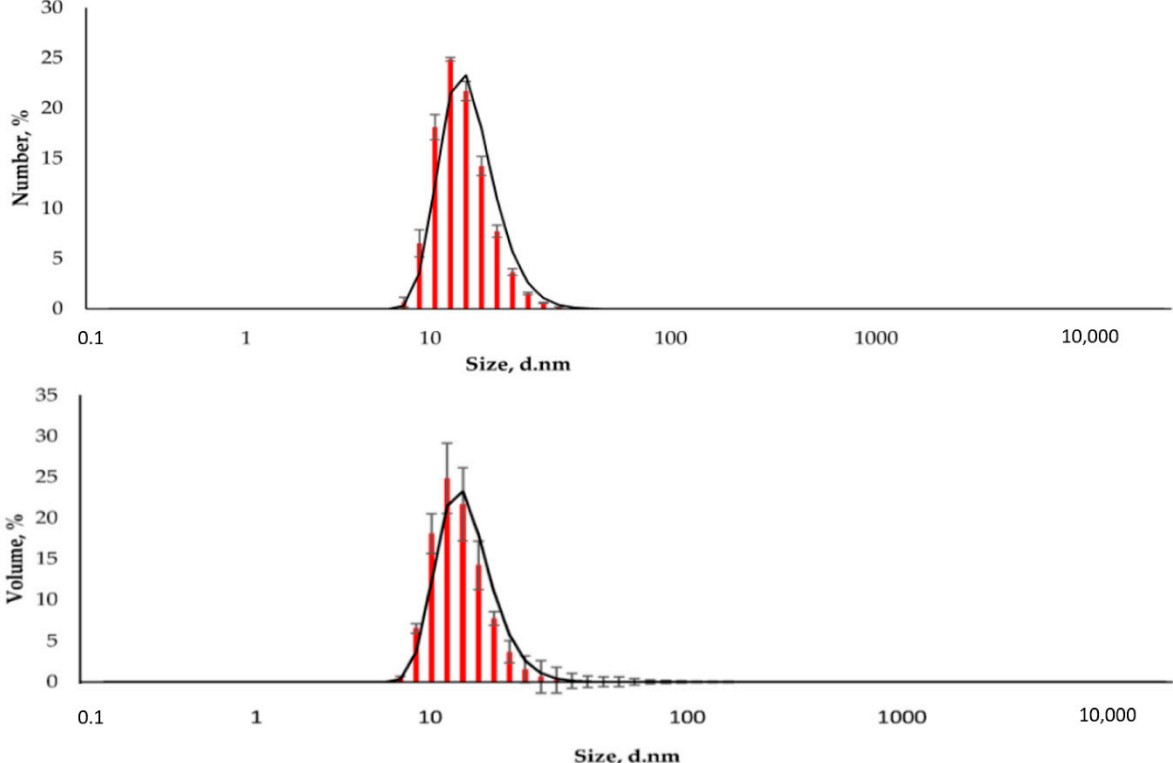

**Figure 3.** The distribution of the hydrodynamic diameter of the nanoparticles was obtained by DLS (Number and Volume distributions).

### 2.2. Toxicity Study of MNPs

Degradation of nanoparticles may lead to the formation of potentially toxic iron ions in the organism. The mechanism of the toxicity includes an imbalance in iron metabolism, reactive oxygen species formation, oxidative DNA damage, and other events [50]. Therefore, the possible toxicity investigation and strategies to avoid any toxic effects are required. One of the widely applied toxicity assays on the cell cultures is the MTT test [51,52]. The viability of cells was investigated using the A549 human lung carcinoma cell line. Lung cancer is a leading cause of cancer death worldwide. Cell cultures were treated with MNP at various concentrations during the exponential growth phase and incubated for 72 h. The experiment is presented in a typical range of MNPs dose, which was used for the in vitro or in vivo investigations [22]. As a result, it was shown that the synthesized MNPs do not have a significant cytotoxic effect at all studied concentrations (Figure 4).

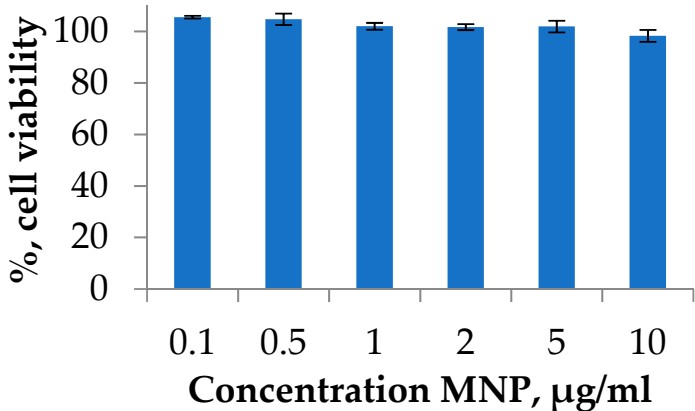

**Figure 4.** Cell viability studies of MNPs using MTT test. A549 cell line was incubated for 72 h with MNPs loaded at various concentrations. Cells treated with PBS buffer were used as a 100% viability control.

MNPs may cause cell death through reactive oxygen species (ROS) formation. This process can be related to the peroxide catalytic degradation by iron ions [21]. ROS production highly depends on the nanoparticles' size, distribution, stability, and surface modification. For this reason, synthesized MNPs were tested in the methylene blue (MB) discoloration assay using frequently used oxidizing agent hydrogen peroxide [53–55]. ROS generated from the reaction of iron ions with hydrogen peroxide will degrade MB into colorless products. To avoid the influence of MB binding on nanoparticles surface, the degradation % was normalized by the reaction of MNPs and MB without $H_2O_2$ (Figure 5). Not normalized data is presented in Supplementary Figure S5. As shown in Figures 5 and S5, the MB was not significantly affected by the MNPs addition.

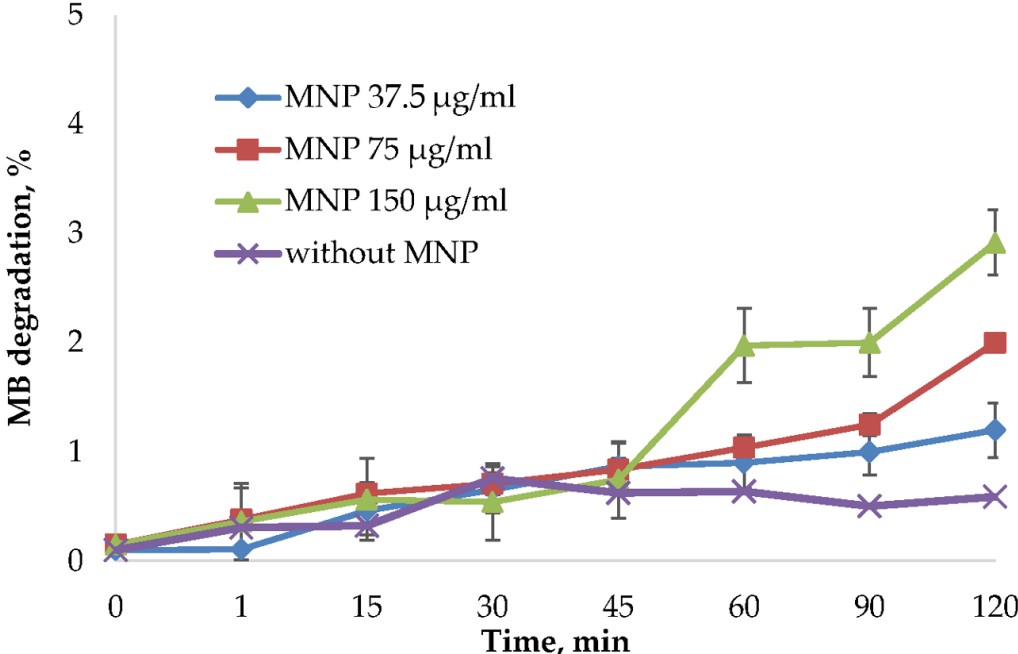

**Figure 5.** MB degradation assay using $H_2O_2$ as an oxidizing agent. The MB degradation % was calculated from the absorbance decrease at 665 nm using a Clariostar plate reader (BMG Labtech, Germany). The reactions were carried out using 37.5 μg/mL, 75 μg/mL, and 150 μg/mL MNP and control solution without nanoparticles. The percentage of dye binding was subtracted from the percentage of degradation of MB in the reaction with MNP and $H_2O_2$.

### 2.3. Analysis of Oligonucleotide Affinity for MNP in Various pH and Buffer Conditions

The affinity of oligonucleotides for MNP was investigated using radioactively labeled oligonucleotide T26 with a fixed amount of MNP (Table 1). The mixture was incubated for 1 h at 25 °C under stirring (see Section 3.8). The oligonucleotide affinity for MNPs was assessed by the radioactivity intensity of the nanoparticles and the solution after magnetic decantation determined using a Vavilov–Cherenkov scintillator. The MNPs were recharged with oligonucleotide to form negatively charged MNPs according to $\zeta$-potential measurement (Table 1). The effect of pH (from 3 to 9.5) and buffer components on the oligonucleotide affinity for MNP was assessed using 10 mM sodium acetic buffer, sodium borate buffer (SBB), phosphate-buffered saline (PBS), and alkaline phosphatase (AP) buffer. As a result, under the low pH range in water and acetic buffer, the highest oligonucleotide affinity for MNP 46–48 nmol/mg was obtained. A decrease in capacity is observed with an increase in the pH of the acetic buffer solution. The positive surface charge of MNP plays a crucial role in negatively charged (due to phosphate groups) oligonucleotide binding. The high capacity at low pH values is probably associated with the higher positive charge of MNP (cf. $\zeta$-potential of MNP in acetic buffer pH 3.0 and 7.0). Surprisingly, in AP buffer with pH 7.5, the efficiency of oligonucleotide interaction with nanoparticles is close to the maximum capacity of 47 nmol/mg (Table 1). The possible mechanism of such an effect may be related to the buffer components such as Tween-20 (polyoxyethylene sorbitol ester), leading to MNP surface stabilization. To study the effect of ionic strength, varying concentrations of aqueous NaCl 10, 50, and 150 mM were used. However, no impact on MNP capacity was obtained.

**Table 1.** Hydrodynamic size, polydispersity index (PDI), and $\zeta$-potential of MNP in various buffer and pH conditions.

| Oligonucleotide | Solution | pH | Hydrodynamic Diameter, nm | PDI | $\zeta$-Potential, mV | Oligonucleotide/MNP nmol/mg |
|---|---|---|---|---|---|---|
| − * | Water | 5.5 | 15.6 ± 2.2 | 0.296 ± 0.003 | 36.0 ± 0.9 | − |
| + | | 5.5 | 180 ± 3 | 0.25 ± 0.01 | −29.0 ± 0.8 | 48 ± 2 |
| − | Acetic | 3.0 | 44 ± 0.5 | 0.29 ± 0.02 | 35 ± 4 | − |
| + | buffer | 3.0 | 133 ± 5 | 0.166 ± 0.007 | −13 ± 3 | 47 ± 2 |
| − | Acetic | 4.0 | 40 ± 0.3 | 0.244 ± 0.009 | 36.0 ± 0.6 | − |
| + | buffer | 4.0 | 116 ± 5 | 0.19 ± 0.1 | −23 ± 1 | 46 ± 2 |
| − | Acetic | 5.0 | 45 ± 0.4 | 0.276 ± 0.006 | 33 ± 1 | − |
| + | buffer | 5.0 | 120 ± 1.5 | 0.254 ± 0.006 | −21.0 ± 1.5 | 16 ± 1 |
| − | Acetic | 6.0 | 46 ± 0.2 | 0.259 ± 0.005 | 25 ± 3 | − |
| + | buffer | 6.0 | 112 ± 2 | 0.33 ± 0.01 | −26.0 ± 0.8 | 19 ± 1 |
| − | Acetic | 7.0 | 63 ± 3 | 0.41 ± 0.01 | 29 ± 2 | − |
| + | buffer | 7.0 | 126 ± 3 | 0.39 ± 0.03 | −33 ± 5 | 26 ± 2 |
| − | PBS | 7.4 | 117 ± 1 | 0.14 ± 0.02 | −36 ± 1 | − |
| + | | 7.4 | 98 ± 1 | 0.132 ± 0.004 | −28 ± 3 | 9.6 ± 0.1 |
| − | AP | 7.5 | 37 ± 1 | 0.223 ± 0.008 | 31.0 ± 0.8 | − |
| + | | 7.5 | 145 ± 2 | 0.22 ± 0.03 | −30 ± 2 | 47 ± 2 |
| − | SBB | 8.3 | 93 ± 1 | 0.21 ± 0.01 | 26 ± 3 | − |
| + | | 8.3 | 172 ± 3 | 0.27 ± 0.02 | −16 ± 0.5 | 9.9 ± 0.1 |
| − | AP | 9.5 | 45 ± 2 | 0.227 ± 0.002 | 31 ± 2 | − |
| + | | 9.5 | 112 ± 1 | 0.185 ± 0.007 | −26 ± 1.4 | 16 ± 1 |

\*—no oligonucleotide.

Significant differences in the oligonucleotide affinity are observed for buffer solutions, AP and PBS with similar pH values (7.5 and 7.4, respectively). Moreover, the oligonucleotide affinity in the PBS buffer is the lowest from obtained one (9.6 nmol/mg). The possible explanation is the change in the surface charge of MNP. $\zeta$-potential of nanoparticles in PBS is −36 mV. This phenomenon is probably associated with the formation of a salt shell that prevents the interaction of the oligonucleotide with nanoparticles. However,

poor oligonucleotide sorption still occurs due to the "displacement effect." To study the effect of phosphate ion on the affinity of the oligonucleotide for MNP, the capacity of MNPs was determined with an increase in the phosphate concentration from 10 to 100 mM. As expected, the capacity decreased by 7.7 times and amounted to $1.3 \pm 0.3$ nmol/mg. The unambiguous effect of the pH and the buffer solution compositions' influence on the efficiency of the interaction of MNPs with oligonucleotides is required for further investigations. Moreover, the PBS and SBB buffers conditions are not optimal for the biological application because the significant aggregation process yields ~100 nm nanoparticles.

### 2.4. Analysis of Various Oligonucleotide Sequence Affinity for MNP

One of the best-obtained capacities for the model oligonucleotide was in the water and acidic conditions. For low pH values, particles might react with the acid. However, the presented nanoparticles were washed at the synthesis stage by acids, and most surface stress points already reacted with acid. To avoid any possible effects, the effect of the oligonucleotide sequence on the affinity for MNP was studied using T26, A26, C26, and X26 oligonucleotides without buffers in water (Table 2). The sequence carrying only guanine residues (G26) was not studied since it can form intra- and intermolecular complexes. The complex formation may have a significant effect on the capacity of nanoparticles. Table 2 shows the values of the MNP capacity with oligonucleotides sequence. The oligonucleotide affinity increases in the series X26 ~ A26 < C26 < T26. However, the effect of the oligonucleotide sequence differs not more than two-fold and is insignificant.

**Table 2.** The comparison of the MNP capacity using various oligonucleotide sequences in an aqueous solution.

| Name | Sequence | Oligonucleotide/MNP nmol/mg |
|------|----------|------------------------------|
| T26 | TTTTTTTTTTTTTTTTTTTTTTTTTT | $48 \pm 2$ |
| A26 | AAAAAAAAAAAAAAAAAAAAAAAAAA | $25 \pm 1$ |
| C26 | CCCCCCCCCCCCCCCCCCCCCCCCCC | $31 \pm 1$ |
| X26 | TTTTTTTCAGGCAGTACCACAAGGCC | $25 \pm 1$ |

## 3. Materials and Methods

### 3.1. Materials

The $FeCl_2 \cdot 4H_2O$ was purchased from Acros organics (MW = 198.81, 99+%). The $FeCl_3 \cdot 6H_2O$ was obtained from PanReac AppliChem (MW = 270.32, 97–102%). MTT (3-[4,5-dimethylthiazol-2-yl]-2,5-diphenyl-tetrazolium bromide) was purchased from Invitrogen. All solvents, reagents were purchased from Sigma (St. Louis, MO, USA) at the highest available grade and used without purification. We prepared the following buffer solutions: 0.4 M sodium acetic buffer, sodium borate buffer (SBB, pH 8.7, 0.05 M $Na_2B_4O_7 \cdot 10H_2O$, 0.1 M HCl), phosphate-buffered saline (PBS, pH 7.4, 137 mM NaCl, 2.7 mM KCl, 10 mM $Na_2HPO_4$, 1.76 mM $NaH_2PO_4$), alkaline phosphatase (AP) buffer (10 mM Tris-HCl, 50 mM KCl, 0.1% Tween-20, 1.8 mM $MgCl_2$) which was adjusted to the pH 7.5 or 9.5 by sodium hydroxide.

### 3.2. MNPs Synthesis

A one-pot co-precipitation method was used to obtain the iron oxide nanoparticles at room temperature [47–49]. Typically, 4 mL of a 1 M aqueous solution of iron (III) chloride, 1 mL of a 2 M aqueous solution of iron (II) chloride, and 10 mL of 2 M HCl were mixed. To the mixture, 50 mL of 0.7 M $NH_3$ (28%) was added and was stirred for 30 min at 500 rpm. The particles were magnetically decanted, the solution was discarded. Then, the precipitate was washed one time with 50 mL of 2M $HClO_4$ and not less than 5 times with deionized water by magnetic decantation. The nanoparticles were then re-suspended in deionized water and stored under nitrogen at 4 °C.

### 3.3. MNPs Characterization

MNPs were characterized using DLS and $\zeta$-potential measurements on a Malvern Zetasizer Nano device (Malvern Instruments, Worcestershire, UK) and TEM on a Jem-1400 device (Jeol, Tokyo, Japan) at an accelerating voltage of 80 kV. For DLS and $\zeta$-potential studies, MNPs were diluted in deionized water to a 300 µg/mL concentration.

### 3.4. Cell Culture and Toxicity Assay (MTT Test)

Tumor cell line from lung adenocarcinoma A549 was cultured in Dulbecco's Minimum Essential Medium (DMEM) supplemented with 10% fetal bovine serum (FBS) (Invitrogen), penicillin (100 units/mL), and streptomycin (100 µg/mL) in a humidified at $37.0 \pm 1.0\,^\circ C$, $5.0 \pm 0.5\%$ $CO_2$ incubator in a humid atmosphere. Exponentially growing cells were plated in a 96-well plate ($2 \times 10^3 \pm 0.5 \times 10^3$ cells per well). After overnight incubation, the cells were treated with media containing MNPs (from 0.1 to 10 µg/mL) and incubated for 72 h at a temperature of $37.0 \pm 1.0\,^\circ C$ in $CO_2$ atmosphere. The inhibition of cell proliferation was investigated using a colorimetric assay based on the cleavage of MTT by mitochondrial dehydrogenases in viable cells. In the process, a blue precipitate of formazan formed [51]. To each well, 200 µL of MTT solution (25 mg/mL in DMEM) was added, and the plates were incubated at $37\,^\circ C$ for 3 h. The medium was removed, and formazan was dissolved in 0.1 mL of dimethyl sulfoxide solution. The absorbance at 570 nm (peak) and 620 nm (baseline) was read using a microplate reader Multiscan EX (Thermo Electron Corporation, Waltham, MA, USA). Results were expressed as a percentage of the control values. All measurements were repeated not less than three times.

### 3.5. Evaluation of Reactive Oxygen Species (ROS) Generation

ROS generation was measured using methylene blue decolorization assay [55] in a 96-well plate at $25\,^\circ C$. The samples (100 µL) were prepared by diluting stock concentrations of methylene blue (MB) to 5 µg/mL, hydrogen peroxide to 0.245 M, and MNPs to 37.5, 75, and 150 µg/mL. The concentration of MB was measured by UV spectroscopy at $\lambda$ 665 nm every 2 min for 80 min using a Clariostar plate reader (BMG Labtech, Ortenberg, Germany). As a control, the reaction of MB with hydrogen peroxide was carried out.

### 3.6. Synthesis and Isolation of Oligonucleotides

Oligonucleotides were synthesized in an ASM-800 synthesizer (Biosset, Novosibirsk, Russia) according to the standard protocol of the 2-cyanoethyl phosphoramidite method. Oligonucleotides were purified by RP-HPLC performed in the Agilent 1200 series chromatograph (Agilent, Santa Clara, CA, USA) on a column ($4.6 \times 150$ mm) containing the Eclipse XDB-C18 sorbent (5 µm) with a 0–90% linear gradient of acetonitrile concentration in 0.02 M triethylammonium acetate solution for 30 min at a flow rate of 1.5 mL/min. The detection of the intensity of optical absorption was carried out at wavelengths 260, 280, 300, and 360 nm. The target product fraction was evaporated in vacuo. Coevaporations removed the bulk of triethylammonium acetate with ethanol. To remove the protecting dimethoxytrityl group, oligonucleotides were treated with 80% acetic acid ($25\,^\circ C$, 5 min). Purified oligonucleotides were concentrated following by precipitation with 2% $LiClO_4$ in acetone, washing with pure acetone, and desiccation under vacuum. As needed, the oligonucleotides were dissolved in deionized water and stored at $-20\,^\circ C$. The concentration of oligonucleotides was determined by UV spectroscopy [56]. UV spectra were recorded on a UV-1800 spectrometer (Shimadzu, Kyoto, Japan).

### 3.7. Synthesis of Radioactively Labeled Oligonucleotides

The preparation of $^{32}P$-labeled oligonucleotides was carried out using the polynucleotide kinase of the phage T4 (5 units), 0.1 mCi g-[$^{32}P$] ATP in a buffer containing 0.05 M Tris-HCl, pH 7.6, 0.01 M $MgCl_2$, 5 mM DTT [57]. The $^{32}P$-labeled oligonucleotides were isolated by 20% polyacrylamide gel electrophoresis (PAGE) and were eluted from the gel by 1 M $LiClO_4$ solution. The oligonucleotide desalting process was performed using flash

reverse phase chromatography. Specific radioactivity was determined using a Mark III scintillation counter (Nuclear-Chicago, Chicago, IL, USA).

### 3.8. Sorption of Oligonucleotides onto the MNP Surface

Suspensions in water or buffers (final volume was 0.1 mL) containing 31.5 μg MNPs and a 15 μM oligonucleotide were incubated for 1 h at 25 °C under stirring (750 rpm), sedimented by magnetic decantation. The MNPs precipitate was washed with 100 μL water, sedimented by magnetic decantation. The supernatant and the washing solution were collected in separate tubes for further measurements. The experiment for T26, A26, C26, and X26 oligonucleotides was presented in water as one of the best conditions for obtaining excellent capacity on the MNPs surface.

## 4. Conclusions

Here, we suggest a fast and convenient method for coating MNPs with DNA. The technique consists of short incubation MNPs (immediately after the synthesis) with oligonucleotides in optimal buffer conditions. This method provides an excellent capacity of MNP concerning oligonucleotide $48 \pm 2$ nmol oligonucleotides/mg magnetic nanoparticles. This approach does not require additional coatings on the surface of nanoparticles, which significantly simplifies the procedure for obtaining MNPs and oligonucleotide-based bioinspired constructions for their further use. We estimated the affinity of oligonucleotides to MNPs, the first toxicity, and ROS generations studies. Under these conditions, oligothymidylates show the highest affinity, while oligocytidylates and a heterogeneous sequence have the lowest affinity. The non-covalent binding of the oligonucleotide to MNPs is a promising approach to obtain cores for nano constructions for the delivery of nucleic acids and gene delivery.

**Supplementary Materials:** The following are available online at https://www.mdpi.com/article/10.3390/magnetochemistry7090128/s1, Figure S1. DLS size distribution of magnetic nanoparticles. The particle size was determined by the DLS method to be $15.57 \pm 2.18$ nm, with a polydispersity index (PDI) of $0.296 \pm 0.03$ and a $\zeta$-potential of $36 \pm 1$ mV. Figure S2. DLS size distribution of magnetic nanoparticles after 90 days of storage. The particle size was determined by the DLS method to be $14.35 \pm 1.29$ nm, with a PDI of $0.307 \pm 0.04$ and a $\zeta$-potential of $36 \pm 1$ mV. Figure S3. DLS size distribution of magnetic nanoparticles after six months of storage. The particle size was determined by the DLS method to be $14.75 \pm 1.47$ nm, with a PDI of $0.297 \pm 0.02$ and a $\zeta$-potential of $36 \pm 1$ mV. Figure S4. DLS volume distribution of fresh magnetic nanoparticles (up), after 90 days (in the middle), after six months (bottom). Figure S5. Methylene blue (MB) degradation assay in the presence and absence of $H_2O_2$ as an oxidizing agent. The MB degradation % was calculated from the absorbance decrease at 665 nm using a Clariostar plate reader (BMG Labtech, Altenberg, Germany). The reactions were carried out using 37.5 μg/mL, 75 μg/mL, and 150 μg/mL MNP.

**Author Contributions:** Data curation and investigation, E.B. and E.D.; conceptualization, E.D. and A.C.; writing—original paper, E.B. and A.C.; funding acquisition, E.D.; project administration, A.C. All authors have read and agreed to the published version of the manuscript.

**Funding:** This study was funded by the Russian Science Foundation (grant no. 21-64-00017); synthesis and characterization of MNPs was funded by the Ministry of Science and Higher Education of the Russian Federation (state registration no. 121031300042-1).

**Acknowledgments:** The authors are grateful to Koval O. A. and Nushtaeva A. A. (ICBFM SB RAS, Novosibirsk) for providing the human lung carcinoma cell line. We thank Poletaeva Yu. E. and Ryabchikova E. I. (ICBFM SB RAS) for fruitful discussion and support with TEM images.

**Conflicts of Interest:** The authors declare no conflict of interest.

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
