# Peer review of "The Effect of pH and Buffer on Oligonucleotide Affinity for Iron Oxide Nanoparticles"

_magnetochemistry, doi:10.3390/magnetochemistry7090128_

Round 1

Reviewer 1 Report

The authors present the application of iron oxide nanoparticles for stable complex formation with oligonucleotides.

This work is interesting and follows recent trends in iron oxide application for various diseases treatment in the future.

Cited work presents the application of the ratio Fe(III):Fe(II) about 2:1 for Fe(III):Fe(II) for Fe3O4 synthesis, while authors of reviewed paper changed the procedure using the 4:1. Proposed ratio 4:1 does not give as a result the Fe3O4 nanoparticles presented in Fig. 1, rather Fe2O3 with lower magnetization. Did authors test magnetic properties showing the saturation of magnetization as well as coercivity Moreover, the authors do not explain the need for acid in washing the product. Indeed, it would be helpful for washing the precipitate in case of Au NPs deposition onto Fe3O4 like mentioning in cited work, while in here such a strong acid would rather dissolve the powder. Why authors washed the precipitate with acid? In pH below 5 Fe3O4 tend to dissolve.

The procedure of oligonucleotides binding is also not described, only mentioned in the introduction. It should be also cited in an experimental part of this paper.

For DLS and zeta potentials studies authors show only the number of NPs in the function of size, while it is recommended to present the volume instead of the number, also in supplementary materials. The measurements show dynamic changes of the tested system, so to have a general overview on the size of NPs in liquid.

According to the ROS studies the application of iron oxide can be also improved. Fe3O4 has high-affinity fo MB dye. MB can easily adsorb onto the NPs surface, so it would be recommended to perform such studies before testing ROS. The decrease of the signal can be affected by the MB adsorption also decolorizing the solution.

How did the authors solve the problem of Fe3O4 dissolution in acidic media while binding oligonucleotides in a buffer having pH 3? Did the authors check the final mass of the product compared with the initial Fe3O4 mass? It is worth t present TGA results comparing pre and post-production of binding.

Author Response

The authors gratefully thank the Referee for the constructive comments and recommendations, which help to improve the readability and quality of the paper. All the comments are addressed accordingly and have been incorporated into the revised manuscript. Detailed responses to the comments and recommendations are as follows. Please note that all the comments are bold-faced, and the authors' reply follows immediately below the comments. The significant changes in the paper text are highlighted in yellow in pdf file.

Cited work presents the application of the ratio Fe(III):Fe(II) about 2:1 for Fe(III):Fe(II) for Fe3O4 synthesis, while authors of reviewed paper changed the procedure using the 4:1. Proposed ratio 4:1 does not give as a result the Fe3O4 nanoparticles presented in Fig. 1, rather Fe2O3 with lower magnetization. Did authors test magnetic properties showing the saturation of magnetization as well as coercivity Moreover, the authors do not explain the need for acid in washing the product. Indeed, it would be helpful for washing the precipitate in case of Au NPs deposition onto Fe3O4 like mentioning in cited work, while here such a strong acid would rather dissolve the powder. Why authors washed the precipitate with acid? In pH below 5 Fe3O4 tend to dissolve.

We apologize for the mistake in the paper concerning the Fe ions ratio. We have changed it in the sentence, «Particles are formed by mixing aqueous solutions of Fe2+ and Fe3+ iron salts in a ratio of 1:2, respectively». In the experimental part, the ratio of the salt has been correct (par. 3.2 Typically, 4 ml of a 1 M aqueous solution of iron (III) chloride, 1 ml of a 2 M aqueous solution of iron (II) chloride). For the iron oxide nanoparticles synthesis, we have used one of the commonly used published procedures [1–3]. The method is suitable for gold coating and for synthesizing polymer-coated nanoparticles for biological applications [2,3]. The acid washing procedure is a standard technique to obtain the optimal charge of the nanoparticle's surface [4]. Positive charges of the colloids can be obtained in an acidic medium by absorption of H3O+ ions. We obtained the same results of the characteristics of the nanoparticles previously published. WE used for nanoparticles quality characteristics and some other methods (TEM, DLS, etc.). Therefore, we are not present in the paper fully characterization of nanoparticles and commonly used washing procedure discussion.

The procedure of oligonucleotides binding is also not described, only mentioned in the introduction. It should be also cited in an experimental part of this paper.

The procedure of oligonucleotides binding is presented in par. 3.8., some information presented in par. 2.3.

For DLS and zeta potentials studies authors show only the number of NPs in the function of size, while it is recommended to present the volume instead of the number, also in supplementary materials. The measurements show dynamic changes of the tested system, so to have a general overview on the size of NPs in liquid.

We have inserted the volume distribution in SI (Figure S4) and Figure 3 of the paper.

According to the ROS studies the application of iron oxide can be also improved. Fe3O4 has high-affinity fo MB dye. MB can easily adsorb onto the NPs surface, so it would be recommended to perform such studies before testing ROS. The decrease of the signal can be affected by the MB adsorption also decolorizing the solution.

We are thankful for catching this confusing wording. We have extensively revised the presented lines in the text. Figure 5 presented a control experiment of MB degradation with H2O2 without MNP. The other reactions are normalized to the MB binding on MNP surface. We have added Figure S5 as initial experimental data. We can see the series of reactions of MNP, MB, and H2O2 and the series of the reaction/binding control of MNP and MB. However, we can see from the picture that the MB signal lowering is very low (only several %), which means that significantly no MB degradation is detected. In this way, it does not matter the mechanism of MB discoloration for the safety applications if it does not occur.

How did the authors solve the problem of Fe3O4 dissolution in acidic media while binding oligonucleotides in a buffer having pH 3? Did the authors check the final mass of the product compared with the initial Fe3O4 mass? It is worth t present TGA results comparing pre and post-production of binding.

Presented nanoparticles were washed with HCl and perchloric acid. Therefore, the possible surface stress points are already reacted with acid. Of course, chemical stability can be an issue for iron oxide nanoparticles at low pH. At low pH =3, the particles might react with the acid. However, we have not used such low pH value conditions for further oligonucleotide loading. We included the following discussion in the text.

  1. Ahmad, T.; Bae, H.; Rhee, I.; Chang, Y.; Jin, S.U.; Hong, S. Gold-coated iron oxide nanoparticles as a T2 agent in magnetic resonance imaging. J. Nanosci. Nanotechnol. 2012, 12, 5132–5137, doi:10.1166/jnn.2012.6368.
  2. Rudakovskaya, P.G.; Gerasimov, V.M.; Metelkina, O.N.; Beloglazkina, E.K.; Zyk, N. V.; Savchenko, A.G.; Shchetinin, I. V.; Salikhov, S. V.; Abakumov, M.A.; Klyachko, N.L.; et al. Synthesis and characterization of PEG-silane functionalized iron oxide(II, III) nanoparticles for biomedical application. Nanotechnologies Russ. 2015, 10, 896–903, doi:10.1134/S1995078015060105.
  3. Rudakovskaya, P.G. Novel bifunctional organic ligands for gold nanoparticles and magnetite modification and hybrid materials on its basis: synthesis, properties, and applications, Moscow state university, 2015.
  4. Massart, R. Preparation of Aqueous Magnetic Liquids in Alkaline and Acidic Media. IEEE Trans. Magn. 2001, 37, 1980–1981.

Reviewer 2 Report

The manuscript submitted for review deals with the preparation of superparamagnetic nanocomposites with an oligonucleotide corona depending on the pH in selected buffers. The thematic focus of the article is current and interesting. The model of SPIONs as carriers of pharmaceuticals or radiopharmaceuticals or only radionuclides for targeted imaging is a well-studied topic. The authors approached the experiments methodically and responsibly. The text is written carefully and appears to be free from obvious errors and inaccuracies. The authors' wording is clear and distinct. The procedures and methods used for the study are clearly described.
The prepared nanoparticles were characterized by TEM and DLS. As important information, I would recommend measuring FT-IR / Ramana evolutionarily XRPD, stating that they actually synthesized the Fe3O4 modification, not some of Fe2O3. This structural information is missing in the work and with a slight change in the reaction conditions, it is possible to obtain a mixed product (both modifications depending on temperature and pH.) At the same time, the authors would prove that the prepared SPIONs are phase pure.

SPIONs were studied as a drug delivery probe for theranostic purposes. It was illustrated on example [223Ra] -SPIONs in Mokhodoeva et al. (doi: 10.1007 / s11051-016-3615-7). The effect of pH on the sorption of 223Ra has been described in the literature. Does authors provide any speciation analysis and calculations for oligonucleotides in relation with acetate or phosphate buffers similar to above mentioned in literature? Analogous studies of the effects of pH on the sorption of radionuclides have also been performed (DOI: 10.1039 / c9ra08953e and DOI: 10.3390 / ma13081915). What is the stability of the composites prepared in this way in plasma or serum? I would like to suggest adding explanations to the questions raised.

I consider the manuscript to be very good and I recommend its acceptance after minor modifications.

Author Response

The authors gratefully thank the Referee for the constructive comments and recommendations, which help to improve the readability and quality of the paper. All the comments are addressed accordingly and have been incorporated into the revised manuscript. Detailed responses to the comments and recommendations are as follows. Please note that all the comments are bold-faced, and the authors' reply follows immediately below the comments. The significant changes in the paper text are highlighted in yellow in pdf file.

The prepared nanoparticles were characterized by TEM and DLS. As important information, I would recommend measuring FT-IR / Ramana evolutionarily XRPD, stating that they actually synthesized the Fe3O4 modification, not some of Fe2O3. This structural information is missing in the work and with a slight change in the reaction conditions, it is possible to obtain a mixed product (both modifications depending on temperature and pH.) At the same time, the authors would prove that the prepared SPIONs are phase pure.

We apologize for the mistake in the paper concerning the Fe ions ratio. We have changed it in the sentence, «Particles are formed by mixing aqueous solutions of Fe2+ and Fe3+ iron salts in a ratio of 1:2, respectively». In the experimental part, the ratio of the salt has been correct (par. 3.2 Typically, 4 ml of a 1 M aqueous solution of iron (III) chloride, 1 ml of a 2 M aqueous solution of iron (II) chloride). For the iron oxide nanoparticles synthesis, we have used one of the commonly used published procedures [1–3]. The method is suitable for gold coating and for synthesizing polymer-coated nanoparticles for biological applications [2,3]. The acid washing procedure is a standard technique to obtain the optimal charge of the nanoparticle's surface [4]. Positive charges of the colloids can be obtained in an acidic medium by absorption of H3O+ ions. We obtained the same results of the characteristics of the nanoparticles previously published. WE used for nanoparticles quality characteristics and some other methods (TEM, DLS, etc.). Therefore, we are not present in the paper fully characterization of nanoparticles and commonly used washing procedure discussion.

SPIONs were studied as a drug delivery probe for theranostic purposes. It was illustrated on example [223Ra] -SPIONs in Mokhodoeva et al. (doi: 10.1007 / s11051-016-3615-7). The effect of pH on the sorption of 223Ra has been described in the literature. Does authors provide any speciation analysis and calculations for oligonucleotides in relation with acetate or phosphate buffers similar to above mentioned in literature? Analogous studies of the effects of pH on the sorption of radionuclides have also been performed (DOI: 10.1039 / c9ra08953e and DOI: 10.3390 / ma13081915).

We have not done any theoretical calculations or modeling of the iron oxide nanoparticles uptake. However, oligonucleotide binding is a much more hard system than Ra2+. Many effects should be considered for such a large molecule, such as several types of oligonucleotide interaction (ionic, hydrophobic, etc.) with iron oxide surface and several possible interactions (one or multipoint interaction). It is hard to have a simple model of the interaction of ions from the buffers and their influence on the capacity of oligonucleotides.

Presented works are exciting and can be cited in the paper as iron oxide nanoparticles applications.

What is the stability of the composites prepared in this way in plasma or serum? I would like to suggest adding explanations to the questions raised.

At the moment, we have done experiments using model oligonucleotides which are not stable for some enzymes in serum. We significantly simplify the procedure for obtaining MNP's and oligonucleotide-based bioinspired constructions for their further use. We estimated the affinity of oligonucleotides to MNPs, the first toxicity, and ROS generations studies. In the future, we are planning to use modified oligonucleotides with good stability in serum.

  1. Ahmad, T.; Bae, H.; Rhee, I.; Chang, Y.; Jin, S.U.; Hong, S. Gold-coated iron oxide nanoparticles as a T2 agent in magnetic resonance imaging. J. Nanosci. Nanotechnol. 2012, 12, 5132–5137, doi:10.1166/jnn.2012.6368.
  2. Rudakovskaya, P.G.; Gerasimov, V.M.; Metelkina, O.N.; Beloglazkina, E.K.; Zyk, N. V.; Savchenko, A.G.; Shchetinin, I. V.; Salikhov, S. V.; Abakumov, M.A.; Klyachko, N.L.; et al. Synthesis and characterization of PEG-silane functionalized iron oxide(II, III) nanoparticles for biomedical application. Nanotechnologies Russ. 2015, 10, 896–903, doi:10.1134/S1995078015060105.
  3. Rudakovskaya, P.G. Novel bifunctional organic ligands for gold nanoparticles and magnetite modification and hybrid materials on its basis: synthesis, properties, and applications, Moscow state university, 2015.

4.             Massart, R. Preparation of Aqueous Magnetic Liquids in Alkaline and Acidic Media. IEEE Trans. Magn. 2001, 37, 1980–1981.

Reviewer 3 Report

In this manuscript, the authors present a procedure for fabricating DNA-coated magnetic nanoparticles. The work based on an accurate description of the synthesis and on adequate characterization techniques aims to provide a simple procedure for obtaining engineered nanoparticles for biomedical use. The presentation is reasonably clear and well written and  the overall results appear solid.

I suggest some points that I believe should be addressed to improve the quality of the manuscript:

1) Evidence of the nanoparticle phase is lacking in the characterization. If no measurements have been made on actual samples, I suggest that you clearly refer to the literature to quantify the amount of magnetite produced. Also, can the coating / solution affect the phase of the particles by inducing oxidation or reduction?

2) Please comment on the hydrodynamic diameter: why is it so different in different solvents even before adding the coating? Are there any aggregation effects?

3) There are some linguistic problems (misspelled words or syntax). Please check lines 58, 99, 106-107

4) The sentence on line 103-105 is unclear.

5) The sentence on line 112-115 is not very clear. Please consider explaining some applications and the corresponding usual concentration.

Author Response

The authors gratefully thank the Referee for the constructive comments and recommendations, which help to improve the readability and quality of the paper. All the comments are addressed accordingly and have been incorporated into the revised manuscript. Detailed responses to the comments and recommendations are as follows. Please note that all the comments are bold-faced, and the authors' reply follows immediately below the comments. The significant changes in the paper text are highlighted in yellow in pdf file.

Evidence of the nanoparticle phase is lacking in the characterization. If no measurements have been made on actual samples, I suggest that you clearly refer to the literature to quantify the amount of magnetite produced. Also, can the coating / solution affect the phase of the particles by inducing oxidation or reduction?

We apologize for the mistake in the paper concerning the Fe ions ratio. We have changed it in the sentence, «Particles are formed by mixing aqueous solutions of Fe2+ and Fe3+ iron salts in a ratio of 1:2, respectively». In the experimental part, the ratio of the salt has been correct (par. 3.2 Typically, 4 ml of a 1 M aqueous solution of iron (III) chloride, 1 ml of a 2 M aqueous solution of iron (II) chloride). For the iron oxide nanoparticles synthesis, we have used one of the commonly used published procedures [1–3]. The method is suitable for gold coating and for synthesizing polymer-coated nanoparticles for biological applications [2,3]. The acid washing procedure is a standard technique to obtain the optimal charge of the nanoparticle's surface [4]. Positive charges of the colloids can be obtained in an acidic medium by absorption of H3O+ ions. We obtained the same results of the characteristics of the nanoparticles previously published. WE used for nanoparticles quality characteristics and some other methods (TEM, DLS, etc.). Therefore, we are not present in the paper fully characterization of nanoparticles and commonly used washing procedure discussion.

Please comment on the hydrodynamic diameter: why is it so different in different solvents even before adding the coating? Are there any aggregation effects?

As mentioned, DLS displays the characteristics of the hydrated particles and compounds in a solvent solution. Moreover, salt and pH values highly influence the surface and hydrodynamic diameter. In this way, we haven't seen any significant aggregation using acetic and AP buffer. However, there is an aggregation process in PBS and SBB buffer. We included following discussion in the text.

There are some linguistic problems (misspelled words or syntax). Please check lines 58, 99, 106-107

The sentence on line 103-105 is unclear.

The sentence on line 112-115 is not very clear. Please consider explaining some applications and the corresponding usual concentration.

We are thankful for catching this confusing wording. We have extensively revised the presented lines in the text.

  1. Ahmad, T.; Bae, H.; Rhee, I.; Chang, Y.; Jin, S.U.; Hong, S. Gold-coated iron oxide nanoparticles as a T2 agent in magnetic resonance imaging. J. Nanosci. Nanotechnol. 2012, 12, 5132–5137, doi:10.1166/jnn.2012.6368.
  2. Rudakovskaya, P.G.; Gerasimov, V.M.; Metelkina, O.N.; Beloglazkina, E.K.; Zyk, N. V.; Savchenko, A.G.; Shchetinin, I. V.; Salikhov, S. V.; Abakumov, M.A.; Klyachko, N.L.; et al. Synthesis and characterization of PEG-silane functionalized iron oxide(II, III) nanoparticles for biomedical application. Nanotechnologies Russ. 2015, 10, 896–903, doi:10.1134/S1995078015060105.
  3. Rudakovskaya, P.G. Novel bifunctional organic ligands for gold nanoparticles and magnetite modification and hybrid materials on its basis: synthesis, properties, and applications, Moscow state university, 2015.
  4. Massart, R. Preparation of Aqueous Magnetic Liquids in Alkaline and Acidic Media. IEEE Trans. Magn. 2001, 37, 1980–1981.

Round 2

Reviewer 1 Report

Thank you for including suggestions and correct the manuscript.